# Long-Term Outcomes of Local Tirofiban Infusion for Intracranial Atherosclerosis-Related Occlusion

**DOI:** 10.3390/brainsci12081089

**Published:** 2022-08-17

**Authors:** Woochan Choi, Yang-Ha Hwang, Yong-Won Kim

**Affiliations:** 1Department of Neurology, Kyungpook National University Hospital, Daegu 41944, Korea; 2Department of Neurology, School of Medicine, Kyungpook National University, Daegu 41944, Korea

**Keywords:** tirofiban, intracranial atherosclerotic stenosis, endovascular reperfusion therapy, long-term outcome, reocclusion

## Abstract

Local tirofiban infusion has been reported as a rescue strategy for intracranial atherosclerotic stenosis (ICAS)-related stroke. However, the long-term outcomes of local tirofiban infusion during endovascular reperfusion therapy (ERT) for ICAS-related stroke are still uncertain. This study aimed to investigate the long-term outcomes of local tirofiban infusion during ERT. We retrospectively analyzed acute patients with ICAS-related stroke who were treated with local tirofiban as a rescue strategy during ERT. The primary outcomes were ischemic stroke, transient ischemic stroke (TIA), and stroke-related death within 30 days. Secondary outcomes included ischemic stroke and TIA beyond 30 days and up to 2 years after ERT in the corresponding treated vessel, symptomatic brain hemorrhage, any stroke, and non-stroke-related death. During a median follow-up of 24.0 months, 12 patients developed an ischemic stroke and TIA (4 within 30 days and 8 afterward). The 1-year risk of stroke and TIA was 9.2% (95% confidence interval, 8.0–18.6%). This study demonstrates that 1-year outcomes of local tirofiban infusion were comparable to the results of intracranial stenting in patients with symptomatic ICAS. Local tirofiban infusion for ICAS-related stroke may be a feasible rescue strategy that can have a bridging role until the maximum effect of antiplatelet agents is achieved.

## 1. Introduction

Intracranial atherosclerotic stenosis (ICAS) is one of the most common causes of stroke among Asian, Black, and Hispanic populations [1]. Acute ischemic stroke related to ICAS is associated with various mechanisms, including in situ thrombotic occlusion, artery-to-artery embolism, hemodynamic compromise, and branch vessel occlusion [2]. In particular, in situ thrombotic occlusion is frequently associated with complicated or failed endovascular therapy procedures. Endothelial irritations during a procedure, atherosclerotic plaque rupture caused by a thrombectomy device, and promoted platelet aggregation could induce reocclusion of the target vessel [3,4].

Tirofiban is a highly selective, short-acting inhibitor of fibrinogen binding to platelet glycoprotein (GP) llb/llla that blocks platelet aggregation [5]. Theoretically, local tirofiban infusion can stabilize thrombogenic conditions, and recent studies have reported the safety and efficacy of local tirofiban infusion as a rescue strategy in patients with ICAS-related large vessel occlusions (LVOs) during endovascular reperfusion therapy (ERT) [4,6,7,8]. However, long-term outcomes of local tirofiban infusion during ERT in ICAS-related LVO are still uncertain. Therefore, we aimed to evaluate the long-term outcomes of local tirofiban infusion during ERT.

## 2. Materials and Methods

### 2.1. Patients

From a prospectively collected institutional stroke registry, we included all patients with ICAS-related LVO who were treated with ERT and local tirofiban between January 2011 and December 2020 (Figure 1). The inclusion criteria were as follows: (1) patients had acute occlusion of the intracranial internal carotid artery (ICA), middle cerebral artery (MCA) M1, MCA M2, or vertebrobasilar artery (VBA); (2) the time from symptom onset to groin puncture was within 24 h; (3) patients were diagnosed with ICAS-related LVO, based on cerebral angiography as the etiology of stroke, and treated with local tirofiban treatment during ERT; (4) patients achieved modified Thrombolysis in Cerebral Ischemia scale (mTICI) grade 2b to 3 for final reperfusion [9]; (5) confirmed ICAS in magnetic resonance angiography (MRA) or computed tomography angiography (CTA) within 7 days after ERT. The exclusion criteria were as follows: (1) treatments with balloon angioplasty or intracranial stenting for ICAS-related LVO; (2) patients had other stroke etiologies, including vasculitis, arterial dissection, or moyamoya disease. The following clinical and radiological data were collected in our institutional registry: age, sex, vascular risk factors, baseline National Institutes of Health Stroke Scale (NIHSS) score, neurological improvement (the difference between pre-ERT and post-ERT NIHSS score), the use of intravenous recombinant tissue plasminogen activator (rtPA), angiographic findings, hemorrhagic complications, and 90-day modified Rankin Scale (mRS). The study protocol was approved by our Institutional Review Board.

### 2.2. Endovascular Therapy

In accordance with the guideline, eligible patients received intravenous thrombolysis with rtPA before ERT [10]. Stent retrieval and contact aspiration thrombectomy were the main primary ERT strategies. If successful reperfusion was achieved but residual stenosis was seen, follow-up angiography was performed 10–30 min after recanalization. If a follow-up angiogram revealed stenosis aggravation, distal flow stagnation, or reocclusion, we performed repetitive ERT and/or rescue treatments, including local tirofiban infusion, balloon angioplasty, and/or stenting. Tirofiban was intra-arterially administered (0.5 mg to 2.0 mg) as a rescue treatment, which was diluted with 8 mL of normal saline and injected approximately at a rate of 1 mL/min [4].

### 2.3. Post-ERT Management and Follow-Up

Non-enhanced brain CT was performed immediately and 12–24 h after ERT to evaluate hemorrhagic complications. If intracranial hemorrhage was excluded on immediate post-ERT brain CT, dual antiplatelet therapy with aspirin and clopidogrel was administered (patients with iv-rtPA: clopidogrel 75 mg/d and aspirin 100 mg/d; patients without iv-rtPA: clopidogrel 300 mg/d and aspirin 300 mg/d at an initial dose, followed by clopidogrel 75 mg/d and aspirin 100 mg/d). In addition to dual antiplatelets, high-intensity statins were also administered. All patients underwent follow-up brain CTA or MRA within 7 days after ERT to evaluate the patency of the target vessel. Neurological evaluation using the NIHSS was measured at admission (pre-ERT, post-ERT) and at day 7. Clinical outcomes were assessed with the mRS at 3 months by certified neurologists or trained nurses. If patients were unable to visit the outpatient department, a structured telephone interview with the patient or guardian was performed. Vascular events during the follow-up period were recorded. Radiologic findings on CTA or MRA were assessed by 2 certified stroke neurologists.

### 2.4. Evaluation of Clinical and Radiological Outcomes

The primary outcomes were ischemic stroke and transient ischemic stroke (TIA) in the corresponding treated vessel, and stroke-related death within 30 days. Secondary outcomes included ischemic stroke and TIA beyond 30 days and up to 2 years after ERT in the corresponding treated vessel, symptomatic brain hemorrhage, any stroke outside of the territory of the treated artery, and non-stroke-related death. For those who performed follow-up angiographic imaging, changes in ICAS were assessed. We defined progression as worsening of stenosis on follow-up CTA/MRA compared with the initial intracranial angiography [11]. Symptomatic reocclusion was defined as reocclusion that was associated with neurologic deterioration. Intracranial hemorrhage (ICH) was assessed by a non-contrast CT scan. ICH was classified based on the European Cooperative Acute Stroke Study III definition [12]. We defined symptomatic brain hemorrhage as any hemorrhage associated with neurologic deterioration.

### 2.5. Statistical Analysis

Chi-square tests or Fisher’s exact tests were used for categorical variables and presented as the count (n) and percentage (%). The Mann–Whitney *U* test was used for continuous variables and presented as medians and interquartile ranges. A binary logistic regression analysis was performed to find predictors for early reocclusion. Variables with *p* < 0.20 in the univariate analysis were included in the binary logistic regression analysis. The value for statistical significance was a *p*-value less than 0.05. The Kaplan–Meier method was used to evaluate the probability of primary clinical outcomes with a 95% confidence interval (CI) at 30 days and 1 year. We used Graphpad Prism 9.3 for the Kaplan–Meier method (GraphPad Software Inc., San Diego, CA, USA). All other statistical analyses were performed using SPSS version 26.0 (IBM, Armonk, NY, USA).

## 3. Results

### 3.1. Baseline Characteristics and Outcomes of ERT

A total of 115 patients were included in this study. Baseline characteristics, stroke risk factors, and outcomes of ERT are summarized in Table 1. The median NIHSS score on admission was 13.0 points (IQR 9-19). The location of ICAS was the intracranial ICA (*n* = 16), MCA M1 (*n* = 71), MCA M2 (*n* = 7), and VBA (*n* = 21). Thirty-six patients (31.3%) received intravenous rtPA. Following ERT, subarachnoid hemorrhage (SAH) occurred in one patient, and there was no symptomatic brain hemorrhage. All patients received dual antiplatelet therapy (aspirin, clopidogrel) after the ERT for at least 3 months. Atrial fibrillation was detected in nine patients who changed from antiplatelet therapy to anticoagulants.

After discharge, 33 patients received dual antiplatelet therapy for more than a year, 21 patients changed to aspirin only, and 38 patients changed to clopidogrel only. In total, 112 patients received atorvastatin medicine (80 mg: 32 patients; 40 mg: 73 patients; 20 mg: 5 patients; 10 mg: 2 patients), except for 3 patients with hepatic failure.

### 3.2. Follow-Up Clinical and Radiological Outcomes

Of 115 patients, 96 patients had a clinical follow-up of over 12 months. Nine patients were lost to follow-up at a median of 3 months after ERT. Long-term clinical and radiological outcomes are presented in Table 2. During a median follow-up of 24.0 (IQR 12.0–25.0) months, four patients (3.5%) experienced primary outcomes (three ischemic stroke and one stroke-related death). Secondary outcomes occurred in 19 patients. Eight patients (7.0%) experienced ischemic stroke or TIA beyond 30 days and up to 2 years (six ischemic stroke and two TIA). Four patients had secondary outcomes within 6 months (three ischemic stroke and one TIA). Two patients had ischemic stroke at 8 months, one TIA at 16 months, and one ischemic stroke at 24 months. Eleven non-stroke-related deaths occurred (seven pneumonia, three acute kidney injury, and one myocardial infarction). Kaplan–Meier analysis showed that the incidence rate of primary outcomes was 3.5% (95 CI, 3.5–24.9%) and the incidence rate of ischemic stroke and TIA was 9.2% (95 Cl, 8.0–18.6%) at 1 year (Figure 2).

One patient (0.9%) had SAH, and six patients (5.2%) had ICH after ERT. None of the patients had a symptomatic brain hemorrhage. All patients underwent follow-up angiographic evaluation (MRA or CTA) within 7 days after ERT, and reocclusion of the treated artery was detected in 14 patients (3 symptomatic, 11 asymptomatic). Fifty-six of one hundred and fifteen patients (48.7%) had follow-up angiographic imaging after 1 month. Fourteen patients (25.0%, 14/56) developed progression of stenosis in the treated vessel at a median follow-up period of 11.5 months (IQR 7.25–15.0). Delayed reocclusion occurred in three patients (5.4%, 3/56), where two reocclusions occurred within 6 months (one asymptomatic, one symptomatic), and one reocclusion (asymptomatic) occurred within 9 months. In binary logistic regression analysis, no neurological improvement (*p* = 0.001) was the sole predictor of early reocclusion (Table 3).

## 4. Discussion

In an acute stroke setting, neurointerventionists who perform ERT might have limited information about the exact mechanism of stroke in patients receiving ERT. They only know the occlusion site and vascular architecture of the patient. However, awareness of possible ICAS during the procedure would affect the options available to the neurointerventionist. One is intracranial balloon angioplasty/stenting. In patients with ICAS, the delivery of a balloon/stent to the target vessel is usually difficult, especially when passing an atherosclerotic vessel and tortuous carotid siphon. Considering that the clinical outcome of patients is affected by the stroke onset to reperfusion time, it would require additional time to prepare this procedure. In addition, sometimes it could be dangerous, due to the different architecture of the intracranial artery, known for its lack of an external elastic layer [13]. On the other hand, the other option, local tirofiban infusion, can be performed rapidly and easily. It just needs delivery of a microcatheter to the target vessel and drug preparation.

There are several infusible GP llb/lllA inhibitors, including abciximab, eptifibatide, and tirofiban. They have a fast onset time within 20 min, which might have a role in bridged platelet inhibition before achieving full antiplatelet action by oral antiplatelet agents [5]. Abciximab is an irreversible antiplatelet and has a relatively long platelet recovery time (up to 48 h). Hemorrhagic complications are of greater concern for abciximab than for tirofiban [14]. Tirofiban and eptifibatide have similar characteristics, including reversible receptor inhibition, a shorter half-life, and a short platelet recovery time (up to 2–4 h) [5]. Although eptifibatide is not available in Korea, Korean stroke guidelines recommend the use of tirofiban during ERT as a rescue therapy in highly selected patients [15].

Several recent studies reported the safety and efficacy of local tirofiban infusion in patients with ICAS-related LVO who received ERT [4,6,7,8]. However, the long-term outcomes of local tirofiban infusion during ERT in ICAS-related LVO remain unclear. We evaluated the long-term outcomes of local tirofiban in patients with ICAS-related LVO. The main findings of this study are as follows: (1) the incidence of primary outcomes at 30 days and the 1-year incidence of stroke/TIA were 3.5% and 9.2%, respectively; (2) the use of local tirofiban during ERT did not increase the risk of symptomatic brain hemorrhage; and (3) most of the identified reocclusions at the target ICAS lesion in this study occurred in an early period, especially within 7 days, and progression of stenosis was seen in approximately 25.0% of patients in follow-up imaging after 1 month.

This study design did not have a control group receiving medical treatment alone or treatment using other EVT strategies, such as balloon angioplasty or intracranial stenting. Therefore, we reviewed and summarized historical data involving various strategies for ICAS-related LVO (Table 4) [16,17,18,19,20]. For medically treated patients, the WASID trial reported a 1-year event rate of 23% in the aspirin group. Other studies on ICAS receiving aggressive medical treatment found event rates of 5.8–9.4% at 30 days and 12.2–15.1% at 1 year [17,18]. For patients with intracranial stenting, the SAMMPRIS and VISSIT trials reported rates of 14.7–24.1% at 30 days and 20.0–36.2% at 1 year, while the WEAVE/WOVEN trials showed a lower event rate than that of other intracranial stenting trials because of best practices with experienced interventionists and careful patient selection [17,18,19,20]. These results are comparable with the overall results of this study, which yielded an event rate of 3.5% at 30 days and 9.2% at 1 year.

In ICAS-related LVO, ERT must focus on the maintenance of reperfusion, as well as fast reperfusion. The pathomechanism of an ICAS-related LVO can be considered as growth of atheroma, rupture of atherosclerotic plaques, and in situ thrombo-occlusion in relation to preexisting stenosis [21]. Considering the process of the ERT procedure, an occluded thrombus can be removed via retrieval by a stent or aspirated by a large-bore catheter. During this procedure, atherosclerotic plaques can be disrupted by retrieval of the stent and repeated passage of thrombectomy devices. These processes might play a role in reperfusion, but they may have a detrimental effect. The rupture of preexisting atherosclerotic plaques and repeated device manipulation can damage the inflamed plaque and induce the release of tissue factors from the endothelium, which leads to local platelet activation, aggregation, and reocclusion [3,4]. Tirofiban directly binds to the ligand-binding pocket of the GP IIb/IIIa receptor, which has a faster onset of antiplatelet action time, within 20 min, compared to the loading dose of aspirin/clopidogrel, which has a plasma half-life of about 2 h [5,22]. By injecting tirofiban directly into the target arterial lesion, low-dose local tirofiban might achieve potentially higher antiplatelet efficacy with a high plasma concentration at the target lesion compared to intravenous administration. 

The additional effect of local tirofiban administration might also be related to improving microcirculatory reperfusion during ERT. Although digital subtraction angiography revealed complete reperfusion, it could not prove effective reperfusion of the microvascular bed. This phenomenon, i.e., “no flow”, might be associated with the existence of a smaller thrombus within the microcirculation [23]. The Chemical Optimization of Cerebral Embolectomy (CHOICE) trial showed improved functional outcomes in the intra-arterial alteplase group over the placebo group, which might result from the improvement of microcirculatory reperfusion by intra-arterial alteplase infusion after successful reperfusion [24]. Theoretically, tirofiban inhibits platelet aggregation and may reduce the formation of microthrombus. Therefore, local tirofiban infusion might also contribute to improving microcirculatory reperfusion. 

In this study, we evaluated long-term radiological outcomes as well as clinical outcomes. In the past, several studies have reported the natural course of ICAS in medically managed patients, and the rate of ICAS progression ranged from 32.5% to 40% during about a 26-month median follow-up period [25,26]. Furthermore, studies on patients with intracranial stenting found in-stent restenosis rates of 17.6–29.7% within 12 months [20,27,28,29]. In this study, reocclusion was found in 14 patients (12.2%) based on follow-up angiographic imaging within 7 days from ERT. In the 56 patients who had additional follow-up angiographic imaging, progression of ICAS was observed in 25.0% (11 progressions, 3 occlusions), during a median follow-up of 11.5 months. These results are better than those of a past medically treated group and similar to those of the intracranial stenting group. We applied a local tirofiban infusion during ERT and aggressive medical treatment including aspirin, clopidogrel, and high-intensity statins after ERT. We hypothesize that tirofiban would maintain the reperfusion of an occluded artery like an intracranial stent until the maximum effect of antiplatelet agents is achieved. We conjecture that this is the main reason these results are better than medical therapy alone. 

Among the 17 identified patients with reocclusion in this study, 14 cases (82.4%) of reocclusion occurred within 7 days. Previous studies have reported that early reocclusion within 24 to 48 h developed in 2.3–7.2% of patients treated with ERT for an acute LVO [30,31,32,33]. Compared to these results, this study revealed a higher early reocclusion rate. We believe this higher early reocclusion rate might be a result of the fact that we strictly included only patients with ICAS-related LVO. ICAS-related LVO was associated with a higher reocclusion rate after ERT than cardioembolic stroke [31,33]. In another study on emergent angioplasty/stenting for ICAS-related LVO, the early reocclusion rate was 13% [25]. In this study, 11 patients did not show any neurological improvements after ERT, and the clinical condition of the remaining 3 patients was aggravated at the time of reocclusion. We found that a lack of neurological improvement (*p* = 0.001) was a predictor of early reocclusion. Thus, we recommend careful examination of patients and follow-up imaging, especially in the early period after the procedure, to detect reocclusion. 

On the other hand, among three patients with delayed reocclusion, two were asymptomatic. Arterial occlusion related to progressive ICAS may allow robust collaterals to develop with the passage of time [26]. Good collateral circulation may offset the potentially detrimental effect of stenosis with diminished flow beyond a stenotic segment, which could lead to favorable outcomes and less recurrence risk [26,34]. Therefore, we speculate that a delay in reocclusion through the administration of local tirofiban during ERT and aggressive medical treatment after ERT might have contributed to collateral development and the lowering of clinical events.

Regarding pharmacological treatments, therapeutic targets can differ depending on the stages. In the acute stage of ICAS, in situ thrombosis and thrombogenic conditions of target ICAS lesions are known to be the main mechanism in emergent ICAS-related stroke [2,4]. Therefore, fast stabilization of the target lesion is a key factor for the maintenance of reperfusion during ERT, and local tirofiban infusion might be a rescue strategy. In the chronic stage, inhibition of atherosclerotic plaque progression and restoration of endothelial function would be important factors [2]. Thus, we applied available high-intensity atorvastatin continuously. All patients started dual antiplatelet therapy after ERT and maintained it for at least 3 months. After this, the antiplatelet strategy was decided depending on the result of a final transfemoral cerebral angiography and the first follow-up angiographic imaging. If the degree of ICAS was stationary, we changed it to a single antiplatelet (aspirin or clopidogrel). If the degree of ICAS was progressive or severe with distal flow compromise, we continued dual antiplatelet therapy. The use of long-term dual antiplatelets was associated with a reduced trend in atherothrombotic events, as well as an increased risk of bleeding complications [35].

There are some limitations to this study. First, this study was a retrospective single-center study, although the data were collected prospectively, so there could be a possibility of selection bias. Although we applied the same ERT protocol, there were some differences in the post-ERT antiplatelet management after discharge, as previously mentioned. Long-term use of dual antiplatelet therapy still lacks evidence. Nine patients were lost to follow-up at a median of 3 months after ERT, and fifty-eight patients did not have follow-up angiographic imaging data after 1 month from ERT. Therefore, these findings should be interpreted cautiously. Second, this was an uncontrolled single-arm study with no control group. We only evaluated local tirofiban infusion for ICAS-related LVO. Therefore, we reviewed previous studies involving various strategies for ICAS-related LVO. Although our results are comparable with past studies, other ERT strategies including balloon angioplasty and intracranial stenting may result in different outcomes. Third, we applied long-term dual antiplatelet therapy, but there were no clear recommendations for this situation in stroke guidelines. Fortunately, symptomatic brain hemorrhage did not occur during the follow-up period. Finally, radiological outcomes should be cautiously interpreted, because 58 patients did not have follow-up angiographic imaging data. 

## 5. Conclusions

This study demonstrates the long-term outcomes of local tirofiban for ICAS-related LVO. One-year outcomes of local tirofiban were comparable to the results of intracranial stenting in patients with ICAS-related LVO. Local tirofiban infusion for ICAS-related LVO in patients may be a feasible rescue or bridging strategy until the maximum effect of antiplatelet agents is achieved and may show potential for better long-term clinical outcomes.

## Figures and Tables

**Figure 1 brainsci-12-01089-f001:**
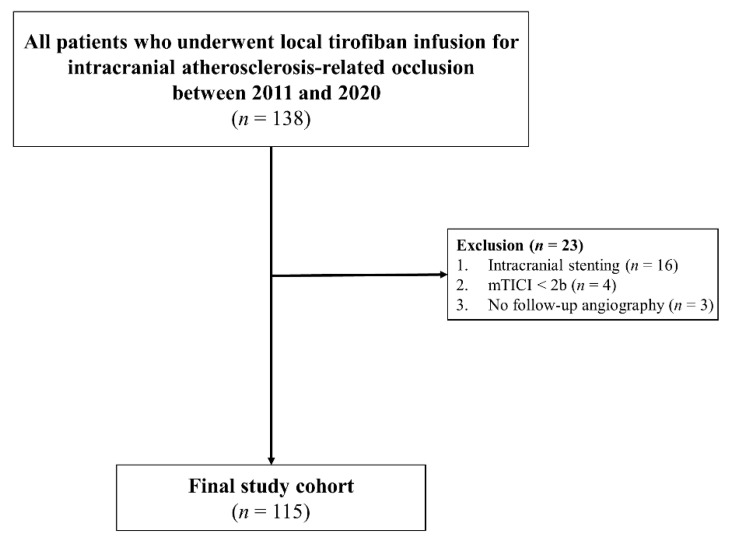
Flow diagram of patient selection. mTICI: modified Thrombolysis in Cerebral Ischemia.

**Figure 2 brainsci-12-01089-f002:**
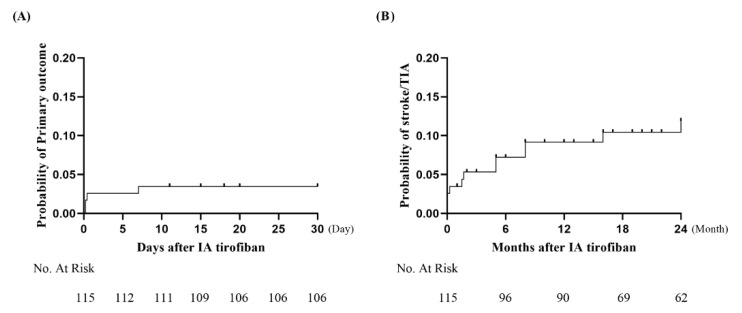
Kaplan–Meier curves in the follow-up period: (**A**) the 30-day rate of primary outcomes was 3.5% (95% Cl, 3.5–24.9%), and (**B**) the 1-year rate of stroke/TIA was 9.2% (95% Cl, 8.0–18.6%). IA, intra-arterial; TIA, transient ischemic attack.

**Table 1 brainsci-12-01089-t001:** Baseline demographics and outcomes of ERT.

Variables	All Patients (*n* = 115)
Age, years	67.0 (59.0–75.0)
Sex, male	77 (67.0%)
Admission NIHSS	13.0 (9.0–19.0)
Intravenous thrombolysis	36 (31.3%)
Occlusion site	
MCA M1	71 (61.7%)
MCA M2	7 (6.1%)
VBA	21 (18.3%)
ICA	16 (13.9%)
Risk factor	
Hypertension	76 (66.1%)
Diabetes	35 (30.4%)
Hyperlipidemia	55 (47.8%)
Smoking	57 (49.6%)
Atrial fibrillation	9 (7.8%)
Coronary artery disease	9 (7.8%)
History of stroke or TIA	22 (19.1%)
Previous antiplatelet	22 (19.1%)
Previous oral anticoagulant	2 (1.7%)
Onset to groin puncture time, min	402.0 (244.0–825.0)
Puncture to final reperfusion time, min	71.0 (52.0–90.0)
Post-ERT intracranial hemorrhage	6 (5.2%)
HI type 1	4 (3.5%)
HI type 2	2 (1.7%)
PH type 1	0 (0.0%)
PH type 2	0 (0.0%)
Post-ERT symptomatic ICH	0 (0.0%)
Post-ERT subarachnoid hemorrhage	1 (0.9%)
mRS score 0–2 at 3 months	66 (57.4%)
Mortality at 3 months	8 (7.0%)

MCA, middle cerebral artery; ICA, internal carotid artery; VBA, vertebrobasilar artery; NIHSS, National Institutes of Health Stroke Scale; TIA, transient ischemic attack; HI, hemorrhagic infarction; PH, parenchymal hemorrhage; ICH, intracranial hemorrhage; min, minutes; ERT, endovascular reperfusion therapy; mRS, modified Rankin Scale.

**Table 2 brainsci-12-01089-t002:** Clinical and radiological outcomes.

Primary Outcome	Patients (*n* = 115)
Ischemic stroke within 30 days	3 (2.6%)
Transient ischemic attack within 30 days	0 (0.0%)
Stroke-related death	1 (0.9%)
Secondary Outcome	
Ischemic stroke beyond 30 days	6 (5.2%)
Transient ischemic attack beyond 30 days	2 (1.8%)
Any stroke outside of treated artery	0 (0.0%)
Symptomatic brain hemorrhage	0 (0.0%)
Non-stroke-related death	11 (9.6%)
Radiological Outcome	
Change in stenosis within 7 days	Patients (*n* = 115)
Stationary	99 (86.1%)
Progression including reocclusion	16 (13.9%)
Reocclusion	14 (12.2%)
Long-term change in stenosis	Patients (*n* = 56)
Stationary	42 (75.0%)
Progression including reocclusion	14 (25.0%)
Reocclusion	3 (5.4%)

**Table 3 brainsci-12-01089-t003:** Univariate and multivariate analysis for early reocclusion.

Variables	Non-Early Reocclusion (*n* = 101)	Early Reocclusion (*n* = 14)	*p* Value	Odds Ratio (95% CI)	*p* Value
Sex, male	70 (69.3%)	7 (50.0%)	0.158	1.802 (0.567–13.539)	0.567
Age, years	67.0 (59.0–75.0)	67.5 (58.8–75.2)	0.952		
Admission NIHSS	13.0 (9.0–19.0)	14 (8.0–20.3)	0.906		
Intravenous thrombolysis	31 (30.7%)	5 (35.7%)	0.705		
Occlusion site			0.545		
MCA	70 (69.3%)	8 (57.1%)			
VBA	17 (16.8%)	4 (28.6%)			
ICA	14 (13.9%)	2 (14.3%)			
Hypertension	69 (68.3%)	7 (50.0%)	0.182	0.69 (0.149–3.204)	0.636
Diabetes	32 (31.7%)	3 (21.4%)	0.439		
Hyperlipidemia	50 (49.5%)	5 (35.7%)	0.338		
Smoking	53 (52.5%)	4 (28.6%)	0.104	2.844 (0.369–21.915)	0.316
Atrial fibrillation	9 (8.9%)	0 (0.0%)	0.999		
Coronary artery disease	8 (7.9%)	1 (7.1%)	0.919		
History of stroke or TIA	19 (18.8%)	3 (21.4%)	0.816		
Previous antiplatelet	20 (19.8%)	2 (14.3%)	0.625		
Previous oral anticoagulant	2 (2.0%)	0	0.999		
No neurologic improvement	17 (16.8%)	11 (78.6%)	0.001	17.907 (3.423–93.694)	0.001
Admission homocysteine, umol/L	11.7 (8.8–14.4)	9.5 (6.5–12.5)	0.055	0.869 (0.702–1.074)	0.194

CI, confidence interval; MCA, middle cerebral artery; ICA, internal carotid artery; VBA, vertebrobasilar artery; NIHSS, National Institutes of Health Stroke Scale; TIA, transient ischemic attack.

**Table 4 brainsci-12-01089-t004:** Comparison of early and delayed outcomes for the current study and other trials.

**Trial**	**Treatment Method**	**30-Day Event Rate**	**1-Year Event Rate**	**Progression of Stenosis**
Current study (*n* = 115)	IA tirofiban, aspirin, clopidogrel, atorvastatin	3.5% (95% Cl, 3.5–24.9%)	9.2% (95% Cl, 8.0–18.6%)	25.0% (14/56)
WASID (≥70% ICAS, *n* = 206) [16], ^‡^	aspirin, statin	N/A	23% (95% CI, 15–30%)	N/A
SAMMPRIS [17], ^§^ (medical arm, *n* = 227)	aspirin, clopidogrel, rosuvastatin	5.8% (95% CI, 3.4–9.7%)	12.2% (95% CI, 8.4–17.6%)	N/A
VISSIT [18], ^||^ (medical arm, *n* = 53)	aspirin, clopidogrel, atorvastatin	9.4% (95% CI, 3.0–20.7%)	15.1% (95% CI, 6.7–27.6%)	N/A
SAMMPRIS [17], ^§^ (WS arm, *n* = 224)	Wingspan stenting, aspirin, clopidogrel, rosuvastatin	14.7% (95% CI, 10.7–20.1%)	20.0% (95% CI, 15.2–26.0%)	N/A
VISSIT [18], ^||^ (WS arm, *n* = 58)	Wingspan stenting, aspirin, clopidogrel, atorvastatin	24.1% (95% CI, 13.9–37.2%)	36.2% (95% CI, 24.0–49.9%)	29.4% (10/34)
WEAVE [19], ^¶^ (*n* = 159) /WOVEN [20], ** (*n* = 129)	Wingspan stenting, aspirin, clopidogrel, statin	2.6% (WEAVE)	8.5% (WOVEN)	17.6% (18/102, WOVEN)

N/A, not available; WASID, Warfarin versus Aspirin for Symptomatic Intracranial Disease; SAMMPRIS, Stenting and Aggressive Medical Management for the Prevention of Recurrent Stroke in Intracranial Stenosis; VISSIT, the Vitesse Intracranial Stent Study for Ischemic Therapy; WEAVE, Wingspan Stent System Post Market Surveillance; WOVEN, Wingspan One-year Vascular Events and Neurologic Outcomes; IA, intra-arterial; WS, Wingspan stenting. ^‡^ Ischemic stroke, brain hemorrhage, or non-stroke vascular death. ^§^ Stroke or death within 30 days after enrollment, or after a revascularization procedure for the qualifying lesion during the follow-up period, or ischemic stroke in the territory of the qualifying artery between day 31 and the end of the follow-up period. ^||^ Any stroke in the same territory within 12 months of randomization, or hard TIA in the same territory between day 2 and month 12 postrandomization. ^¶^ Stroke or death within 72 h. ** Stroke within the target artery territory, non-traumatic hemorrhage, or neurologic death within 1 year following stenting.

## Data Availability

Not applicable.

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
