# Peer review of "Long-Term Outcomes of Local Tirofiban Infusion for Intracranial Atherosclerosis-Related Occlusion"

_brainsci, 2022, doi:10.3390/brainsci12081089_

Round 1
Reviewer 1 Report
This paper "Long-term outcomes of local tirofiban infusion for intracranial 2 atherosclerosis-related occlusion" it is an excelent work, original where it has aligned the objetives, Matherial and methods, results and conclusiones. Use up to date bibliography. Only that in table 1 and the end of the age I would add "years"
Author Response
We appreciated the reviewer’s comment. We have updated references No. 2 and No. 14. And we added “years” at the end of the age in Table 1.

Reviewer 2 Report
To:
Editorial Board
Brain Sciences
Title: “Long-term outcomes of local tirofiban infusion for intracranial atherosclerosis-related occlusion”
Dear Editor,
I read this paper and I think that:
- Abbreviations should be expressed at their first mention both in the abstract and in the main text. Please revise the entire manuscript.
- The retrospective nature of this paper might be considered as a limitation. This should be discussed in a dedicated limitation section.
- The impact of baseline pharmacological treatments as well as modifications of therapies after discharge might impact on results. Please discuss such a point and provide detailed data if possible, otherwise this should be considered as a great limitation of the paper.
- All comorbidities of patients should be described and discussed as they can impact on results. Multivariate regression section should take into account this aspect. Please revise the text and the statistical analysis.
Author Response
- Abbreviations should be expressed at their first mention both in the abstract and in the main text. Please revise the entire manuscript.
Response: As you commented, we revised all abbreviations in the abstract and main text.
- The retrospective nature of this paper might be considered as a limitation. This should be discussed in a dedicated limitation section.
Response: The authors completely agreed with your comments. This study was a retrospective analysis of a prospectively enrolled registry, which may have led to bias. Although we applied the same endovascular reperfusion therapy (ERT) protocol described in the Endovascular Therapy section (lines 73-82), there were some differences in the post-ERT antiplatelet management after 3 months from ERT (lines 132-138). And, 9 patients were lost to follow-up at a median of 3 months after ERT, and 58 patients did not have follow-up angiographic imaging after 1 month from ERT. It might be a limitation of this study. We added this content in the limitation section of the discussion. Please refer to lines 301-308.
- The impact of baseline pharmacological treatments as well as modifications of therapies after discharge might impact on results. Please discuss such a point and provide detailed data if possible, otherwise this should be considered as a great limitation of the paper.
Response: The authors agreed with your critical comment. Regarding pharmacological treatments, therapeutic targets can differ depending on the stages. In the acute stage of intracranial atherosclerotic stenosis (ICAS), in situ thrombosis and thrombogenic conditions of target ICAS lesions are known to be the main mechanism in emergent ICAS-related stroke [1-3]. Therefore, fast stabilization of the target lesion is a key factor for maintenance of reperfusion during ERT, and local tirofiban infusion might be a rescue strategy. In the chronic stage, inhibition of atherosclerotic plaque progression and restoration of endothelial function would be important factors [1]. Thus, we applied available high-intensity atorvastatin continuously. All patients started dual antiplatelet therapy after ERT and maintained it for at least 3 months. After then, the antiplatelet strategy was decided depending on the result of between final transfemoral cerebral angiography and the first follow-up angiographic imaging. If the degree of ICAS was stationary, we changed it to a single antiplatelet (aspirin or clopidogrel). If the degree of ICAS was progressive or severe with distal flow compromise, we continued dual antiplatelet therapy. We added this content in the Discussion (lines 286-300) and limitations (lines 303-305). Detailed data was described in the Results section (lines 132-138).
References
- Kim, J.S.; Bang, O.Y. Medical Treatment of Intracranial Atherosclerosis: An Update. J Stroke 2017, 19, 261-270, doi:10.5853/jos.2017.01830.
- Kang, D.H.; Kim, Y.W.; Hwang, Y.H.; Park, S.P.; Kim, Y.S.; Baik, S.K. Instant reocclusion following mechanical thrombectomy of in situ thromboocclusion and the role of low-dose intra-arterial tirofiban. Cerebrovasc Dis 2014, 37, 350–355
- Kim, Y.W.; Sohn, S.I.; Yoo, J.; Hong, J.H.; Kim, C.H.; Kang, D.H.; Kim, Y.S.; Lee, S.J.; Hong, J.M.; Choi, J.W.; et al. Local tirofiban infusion for remnant stenosis in large vessel occlusion: tirofiban ASSIST study. BMC Neurol 2020, 20, 284
- All comorbidities of patients should be described and discussed as they can impact on results. Multivariate regression section should take into account this aspect. Please revise the text and the statistical analysis.
Response: As we described in the statistical analysis section, we performed univariate analysis using all comorbidities of patients, and variables with p<0.20 in the univariate analysis were included in the multivariate regression analysis. We revised table 3.
We found that a lack of neurological improvement (p=0.001) was a predictor of early reocclusion. Thus, we recommend careful examination of patients and follow-up imaging, especially in the early period after the procedure, to detect reocclusion. Please refer to lines 275-277.

Reviewer 3 Report
the paper is good and well written.
Author Response
We appreciate the reviewer’s favorable comment.

Round 2
Reviewer 2 Report
Authors well addressed my previous comments. THe paper improved very much